# Characterization of β-Glucan-Peanut Protein Isolate/Soy Protein Isolate Conjugates and Their Application on Low-Fat Sausage

**DOI:** 10.3390/molecules27093037

**Published:** 2022-05-09

**Authors:** Manli Zhang, Hongzhi Liu, Qiang Wang

**Affiliations:** Key Laboratory of Agro-Products Processing, Ministry of Agriculture and Rural Affairs, Institute of Food Science and Technology, Chinese Academy of Agricultural Science, Beijing 100081, China; manlizhang165@163.com

**Keywords:** polysaccharide-protein conjugates, emulsifying properties, structure, low-fat sausage

## Abstract

Polysaccharide–protein conjugates can improve the functional properties and expand the application field. The emulsifying, thermal properties of WSG-PPI conjugates and WSG-SPI conjugates were improved, compared to WSG, PPI and SPI. The Maillard reaction was confirmed by Fourier transform infrared spectroscopy (FT-IR). Circular dichroism (CD) exhibited that the structure of the conjugates was more expanded. Cryo-SEM and AFM demonstrated that the WSG, WSG-PPI and WSG-SPI conjugates had a morphology of a chain. When the conjugates were added as fat substitutes to low-fat sausage, the cooking yield, hardness and chewiness increased. The objective of this research was to study the emulsifying property, thermal property and structural changes of β-glucan-peanut protein isolate (WSG-PPI) conjugates and β-glucan-soy protein isolate (WSG-SPI) conjugates prepared through wet-heated Maillard reaction, and their effect on the texture of low-fat sausage.

## 1. Introduction

*Saccharomyces cerecisiae* β-glucan is a kind of polysaccharide, which is linked by a 1→3 glycosidic bond as the main chain and a 1→6 glycosidic bond as the branch chain [1,2]. It is present in the cell wall, which can contribute to the integrity of the cell wall, cell exchanges, and keep cells from pathogens and environmental stresses [3]. It has superior bioactivities, such as anti-tumor [4,5,6,7,8], antioxidant [9,10], hypoglycemic effect [11], immune regulation [12,13] and antibacterial effects [7]. However, β-glucan is hardly soluble in water and common solvents owing to its compact triple helix conformation [14]. It is difficult for β-glucan to react with protein in an aqueous solution. In this paper, a safe, efficient, and green enzymatic hydrolysis method was selected to prepare water-soluble β-glucan (WSG). WSG was the product obtained by the enzymatic hydrolysis of β-glucan. It was more soluble in water with a reduced molecular weight and had a highly ordered structure. Previous studies proved that WSG has an emulsifying property, antioxidant property [15,16], and that it stimulates immunity [17,18], which can be used in food, medicine, cosmetics and other fields [19].

The Maillard reaction is one of the major non-enzymatic browning reactions in food processes, which improves the functional properties of protein or polysaccharides [20,21,22]. For example, soy hull hemicelluloses-soy protein isolate conjugates demonstrated substantially better emulsification capacity and thermal stability than soy protein [23]. Myofibrillar protein-dextran conjugates exhibited emulsifying property and solubility [24]. Whey protein isolate-gum acacia possessed an improved emulsifying property, high solubility and stability heat-induced insolubility, and reduced surface hydrophobicity [25]. Oat β-glucan-dipeptide exhibited improved an apparent viscosity, emulsifying property, swelling power and fat binding capacity [26]. The previous studies were mainly about the functional properties and structures of protein during a Maillard reaction. Polysaccharides, as a substrate of the Maillard reaction, widely exist in plants. However, there are few studies on the changes of physicochemical properties of polysaccharides after a Maillard reaction.

Sausage is one of the most popular meat products in the world, and has a high content of fat, with an unhealthy fatty acid profile. However, with the improvement of quality of life, people pay more and more attention to the relationship between diet and health. Consumers’ purchasing habits are changing, and low-fat food becomes more and more popular. Fat plays an important role in maintaining the sensory characteristics such as texture and juiciness of the meat product. The direct reduction of fat may lead to the poor quality of the sausage, as many ingredients had been studied as fat replacers to decline fat levels without compromising sausage quality. Pork skin-based emulsion gels elaborated with canola oil, bamboo fiber and inulin were promising alternatives to replace pork back fat in low-fat sausages [27]. Emulsion gels prepared with carrageenan, and emulsion stabilized by zein/carboxymethyl dextrin also served as a fat substitute in sausages, and could enhance the hardness and viscoelasticity of pork sausage [28]. The addition of 15% chicken skin, wheat fiber mixture or pineapple dietary fibers to sausage could significantly reduce the cooking yield [29,30]. The effect of pea protein isolate, pea low moisture extrudate and pea high moisture extrudate replacing 20% pork meat were compared; there are no significant differences between sausage with PPI and normal sausage, and the sausage with pea extrudate had a softer bite and significant color changes [31].

The reaction between WSG and amino acid/protein was inevitable in the process of food processing. The purpose of this paper was to study the functional properties and structural changes of WSG and peanut protein isolate (PPI) and soy protein isolate (SPI) during Maillard reaction, as well as the effect of WSG-PPI conjugates and WSG-SPI conjugates on the texture of low-fat sausage.

## 2. Results and Discussion

### 2.1. Functional Properties

#### 2.1.1. Emulsifying Properties

Emulsifying properties were related to the spatial conformation and physico-chemical properties of the substance itself. The conjugates bound the polyhydroxy carbohydrate to protein through the Maillard reaction, and the protein exposed a certain hydrophobic group to quickly and tightly absorb on the oil-water interface [32]. Otherwise, the formation of conjugates was affected by many factors, and then affected the emulsification.

The emulsifying activity index (EAI) and emulsifying stability index (ESI) of different conjugates were compared (Figure 1). The EAI and ESI of all conjugates were higher than the EAI and ESI of the WSG. There were two reasons to explain the results. First, the access of hydrophobic groups allowed the conjugates to be more effectively absorbed to the oil–water interface by the reaction between protein with WSG [33]. Second, WSG could form a steric repulsion between the surface of emulsion particles and promote the formation of a stable membrane around the oil particles [34]. Moreover, the second structure of the molecules was extended by heat treatment [35]. Chen et al. [25] also reported that the emulsifying properties of WPI-GA conjugates were improved compared to the WPI and WPI-GA mixture. Sugar beet pectin-whey protein isolate/bovine serum albumin conjugates crosslinked by Genipin exhibited excellent emulsifying properties [36].

However, there were no significant differences between the emulsifying properties of WSG-PPI, WSG-arachin and WSG-conarachin conjugates. The same results were observed between WSG-SPI and WSG-7S globulin conjugates. Then, the WSG-PPI and WSG-SPI conjugates were selected for further study.

#### 2.1.2. Contact Angle and Particle Size Distribution

The amphiphilicity and particle size distribution were factors affecting emulsion stability. The wettability of the particles directly affected its adsorption on the oil–water interface [37]. The smaller the angle, the better the wettability. If the angle was <90°, the solid surface was hydrophilic; conversely, the solid surface was hydrophobic. The contact angle was used to measure the amphiphilicity and wettability of PPI, SPI, WSG, WSG-PPI and WSG-SPI conjugates (Figure 2). The contact angle of WSG, PPI, SPI, WSG-PPI and WSG-SPI conjugates was 50.1°, 101.9°, 117.3°, 94.1° and 85.9°. The contact angle of conjugates was higher than WSG, which was due to the increase in hydrophobic groups in the polysaccharide’s chain. However, the contact angle of conjugates was lower than SPI and PPI; it could be inferred that the hydrophilic groups in the protein molecules increased due to the combination of the polysaccharide chain and protein chain. 

The mean particle size reflected the aggregation behavior of WSG-PPI and WSG-SPI conjugates. The mean particle of WSG, WSG-PPI and WSG-SPI was 38.25 ± 0.17 μm, 50.07 ± 0.76 μm and 104.89 ± 3.79 μm, respectively. The mean particle size of WSG-PPI and WSG-SPI conjugates increased compared to the WSG and WSG-PPI and WSG-SPI mixture (Figure 3a). The particle size of the polysaccharide was related to its molecular weight, intrinsic viscosity, and physical stability [38]. WSG and PPI or SPI formed larger molecules through the Maillard reaction, which might explain the increase in particle size. Patel found that the particle size of the bio-polymer was larger [39].

#### 2.1.3. Thermal Properties

The thermal property of conjugates could be studied by measuring the temperature and rate of pyrolysis. There were two thermal degradation process during the 30–500 °C heating treatment (Figure 3b). The initial weight loss was due to the evaporation of free and bound water during 30–150 °C. The second weight loss was attributed to the breaking of the intermolecular and intramolecular hydrogen bonds and electrostatic bonds, and the decomposition of the hydrophobic interaction during the temperature of 150–500 °C. WSG, PPI, SPI and WSG-PPI and WSG-SPI conjugates began to decompose drastically and substantially at 283.2 °C, 310.1 °C, 312.9 °C and 277.1 °C and 306.4 °C, respectively. However, the weight loss of WSG, PPI, SPI and WSG-PPI and WSG-SPI conjugates was 68.5%, 67.5%, 61.3% and 32.4% and 64.9%, respectively. It was inferred that thermal stability of WSG-PPI conjugates was improved compared to WSG and PPI, and the thermal stability of the WSG-SPI conjugates was lower than SPI, which was higher than WSG. The thermal stability of conjugates was related to the type of protein and the ratio of protein to polysaccharide. The thermal stability of caseinate-maltose conjugates was lower than caseinate [40]. The thermal stability of carboxymethyl cellulose(CMC)/SPI blend films improved due to the addition of the CMC content [41]. The improvement was also attributed to the Maillard reaction and the crosslinking between SPI and carboxymethyl cellulose. The thermal stability of soy hemicelluloses-SPI conjugates (1:9) was lower than conjugates with more SPI content [23]. Schizophyllan modified by α-amino also demonstrated better thermal stability compared to the unmodified schizophllan +poly(c) complex [42]. 

#### 2.1.4. Rheological Properties


(1)Viscosity


The viscosity of polysaccharide was affected by its molecular weight. The increase in molecular weight can make the molecular chain extend and the side chain increase, making it easier to become entangle between the molecules. The viscosity profiles of WSG and its conjugates were presented in Figure 3c. WSG and its conjugates demonstrated shear thinning behavior at 25 °C, which was also observed in oat β-glucan and its conjugates [43]. This behavior was due to the distribution of random coil polymers or their parallel alignment with the flow stream [44]. In addition, the viscosity of WSG-PPI and WSG-SPI conjugates apparently increased. The change of viscosity might be attributed to the change of structure, molecular weight and concentration of samples [26]. The molecular weight of WSG-PPI and WSG-SPI conjugates was higher than WSG′s molecular weight by the Maillard reaction; then, the viscosity of the WSG-PPI and WSG-SPI conjugates increased. Su et al. [41] also found that the reaction between soy protein isolate and carboxymethyl cellulose could cause an increase in viscosity. 


(2)Frequency sweep test.


The storage modulus (G′) and the loss modulus (G″) represented the elastic property and the viscous property in the frequency sweep test, respectively. The G′ and G″ of WSG and its conjugates were shown in Figure 3c. The intersection between G′ and G″ of WSG indicated the transition from gelation to solation. The G′ of WSG-PPI conjugates and WSG-SPI conjugates was higher than the G″, which indicated a predominant solid-like behavior of the conjugates. The results indicated that the viscoelastic behaviors of WSG changed during the Maillard reaction. G′ and G″ of WSG and its conjugates increased after the Maillard reaction during the whole frequency range. The increase in G′ may be due to the strong molecular interaction stabilizing the closed-pack molecules system [45]. Sun et al. [43] found that the G′ of β-glucan conjugates was related to the reaction substrate, and the G′of β-glucan-amino acid conjugates was higher than β-glucan, but the β-glucan peptide conjugates was similar to β-glucan.

### 2.2. Structure

#### 2.2.1. Fourier Transform Infrared Spectroscopy (FT-IR)

FT-IR was a method to study the conformational changes of WSG and its conjugates in the dry powder state because the chemical fingerprints of polysaccharides and protein did not overlap [46]. The absorption band at the 3400 cm^−1^ corresponded to the stretching vibration of –OH in the constituent sugar residues [47]. The absorption band at 2920 cm^−1^ was due to the stretching vibration of –CH [48]. The absorption band at 1630 cm^−1^ was attributed to the stretching vibration of –CHO and –C=O [49]. The absorption band at 1030 cm^−1^ was associated with the stretching vibration of -C-O-C from the different units of (1→3) and (1→6) in WSG. However, the peak of WSG-PPI conjugates and WSG-SPI conjugates at 1630 cm^−1^ was lower than the peak of WSG, which might be explained by the fact that the Maillard reaction consumed the C=O and –CHO of WSG. The phenomenon also appeared in oat β-glucan-L-glutamic and β-glucan-collagen peptide I conjugates [26]. When the feruloyl groups in pectin molecule were hydrolyzed, a spectral change in the range of 1616–1634 cm^−1^ was observed, which was attributed to the esterified [50].

#### 2.2.2. Circular Dichroism (CD)

CD was an effective method to study the structural changes of protein and polysaccharide. The conformational transition of samples was detected on the CD spectra (Figure 4b). All samples had a positive and negative cotton effect, which suggested that the samples were in a highly ordered structure [51]. However, the ellipticity and maximum/minimum peak position of the samples were different. The CD spectra of WSG had a positive cotton effect with a maximum absorption at 190.0 nm and a negative cotton effect with a minimum absorption at 210.2 nm among 190–250 nm. The maximum absorption of WSG-PPI mixture decreased and shifted to 192.0 nm. The maximum absorption of the WSG-SPI mixture and WSG-SPI conjugates did not change significantly. The minimum absorption peak of the WSG-PPI mixture and WSG-SPI mixture shifted at 208.2 nm and 206.6 nm, respectively. The maximum absorption peak of WSG-PPI conjugates shifted to 191.4 nm, which was lower than the WSG and its mixture. The minimum absorption peak of WSG-PPI conjugates and WSG-SPI conjugates appeared at 205.4 nm and 200.4 nm, remarkably indicating a conformational transformation among the WSG, WSG-PPI/SPI mixture and WSG-PPI/SPI conjugates. The difference between the WSGs and their conjugates’ spectra demonstrated a symmetrical peak at 205 nm, consisting with the nπ* absorbance maximum of esters [52]. β-lg-carboxymethyldextran conjugates were also found in the decrease in the peak in the presence of the polysaccharide, as well as a blue shift [53].

#### 2.2.3. Scanning Electron Microscope (SEM)

The microstructure of WSG and its conjugates in solid state can be observed. WSG presented an irregular slice shape, the WSG-PPI conjugates and WSG-SPI conjugates exhibited aggregated clump shapes with small particles on the surface, and the surface was more uneven (Figure 5), which was due to WSG covalently bonded to PPI or SPI and improving the content of the hydrophilic radical on the surface of PPI and SPI [54]. Otherwise, the heating treatment expended the contact area between PPI/SPI and WSG, leading to the coarse surface of the WSG-PPI and WSG-SPI conjugates. The finding was consistent with Yu [54], who found that the lactose-high-temperature peanut protein isolate conjugates’ surface appeared with an uneven bulge and formed a sugar-surrounding protein structure. The high-temperature peanut protein isolate-sesbania gum conjugate also exhibited a rougher surface [55]. In general, the graft polysaccharide surface tended to be rough and non-uniform, which was favorable for the formation of a more hydrophilic protein surface structure and for dispersion in a water solution [56].

#### 2.2.4. Cryo-SEM and AFM

Cryo-SEM could observe the frozen liquid samples on the pore scale. The sample was frozen at a low temperature and stored in the microscope, which made the sample retain moisture in a high vacuum and allowed the sample to preserve the original structure [57]. The microstructure of WSG, PPI, SPI and its conjugates in solution was observed by cryo-SEM (Figure 6). WSG demonstrated irregular chains, and WSG-PPI conjugates and WSG-SPI conjugates demonstrated an irregular chain with a spherical structure attached. PPI or SPI observed the spherical structure, so it could be inferred that the spherical structure of WSG-PPI and WSG-SPI conjugates chain was PPI or SPI.

AFM was a technique that could characterize biopolymers on sub-nanometer scale, which had the advantages of visualizing the specimen by contouring the forces between the probe and the specimen surface [58]. As AFM figures demonstrated, the WSG exhibited a morphology of a chain and WSG-PPI conjugates and WSG-SPI conjugates exhibited a chain structure with some particles attached, while the PPI and SPI showed small particles.

### 2.3. Application

#### 2.3.1. Cooking Yield of Sausage

The cooking yield of low-fat sausages increased first and decreased as the additional weight of the WSG-PPI conjugates increased, comparing with the normal-fat (NF) sausage (Figure 7a). However, the cooking yield of the low-fat sausage-added WSG-SPI conjugates was slightly lower than the NF sausage. The WSG-PPI conjugate had good emulsifying properties, making the sausage compact and homogeneous. In addition, the WSG was a kind of macromolecular polysaccharide, which could form a network structure with protein. However, the cooking yield of WSG-SPI conjugates decreased, which indicated that the ability of WSG-SPI conjugates to strengthen the meat mixture was less than that of WSG-PPI conjugates. A similar result was found in sausages containing pineapple dietary fiber 60 [30]. Pig skin and wheat fiber mixture as a fat replacer for frankfurter-type sausages could improve their cooking yield and water-holding capability [59]. Low-fat sausage-added oatmeal or tofu increased water retention, and produced less cooking loss [60]. Polysaccharides had the capacity to improve fat and water retention in sausages [61,62]. The effects of CMC on the water-holding capacity of three batters with different viscosities were compared. CMC could improve the water retention of medium viscous batters and high viscous batters, but had no effect on low viscous batters [63].

#### 2.3.2. Water Loss of Sausages

Low-fat sausages lose less water than the NF sausage (Figure 7b). The water-holding capacity of low-fat sausages was increased as a result of the addition of the WSG-PPI and WSG-SPI conjugates. The effect of WSG-PPI conjugates and WSG-SPI conjugates on the water-holding of sausages contained mainly two aspects: for one thing, the hydroxyl groups can absorb water through hydrogen bonds and hydrophobic interactions when the WSG-PPI conjugates and WSG-SPI conjugates were heated [64]; for another, the more compact gel structure could be formed between protein and WSG-PPI conjugates or WSG-SPI conjugates, which wrapped water inside the gel to prevent it from losing [65,66].

#### 2.3.3. Color of Sausages

Color was an intuitive indicator of sausages, and it affected the choice of consumers. As shown in Figure 7c, the lightness (L*), redness (a*) and yellowness (b*) of the low-fat sausages were different to the NF sausage. WSG-PPI and WSG-SPI conjugates caused a decrease in lightness (L*) values. The redness (a*) values increased with the WSG-PPI and WSG-SPI conjugates’ addition. The yellowness (b*) values were higher than the NF sausages. The L* values of the meat product decreased as a decrease in the fat level, then the value of a* was the opposite [67,68]. Câmara et al. [69] reported that the values of L*, a* decreased and the value of b* increased, when chia (*Salvia hispanica* L.) mucilage was added to the meat model system.

#### 2.3.4. Texture Profile Analysis (TPA)

Fat content played a basic role in the texture, flavor, mouthfeel and bite [70]. Fat improved the water-binding capacity and stabilized the gel network of the protein-emulsified meat products [69]. The hardness of low-fat sausages was higher than NF sausages; the highest hardness was 6945.50 ± 460.21, 7045.78 ± 237.79, respectively, when the sausages with WSG-PPI or WSG-SPI replaced 20% fat (Figure 8). The increase in the hardness and chewiness was attributed to the addition of the WSG-PPI and WSSG-SPI conjugates, such as high-water retention capacity, high-fiber content, viscosity and viscoelastic properties, which increased the hardness [71,72]. The result agreed with Choe [29], who found that the addition of chicken skin and wheat fiber led to higher hardness in the sausage. The changes of cohesiveness were consistent with the result of Abbasi [73], who found that 0.5% gum tragacanth increased the cohesiveness of the sausage, while 1% gum tragacanth reduced the cohesiveness, which was related to the fat content and the water-holding capacity of the gum tragacanth.

## 3. Materials and Methods

### 3.1. Materials

*S. cerecisiae* β-glucan was purchased from Angel Yeast Co., Ltd. (Hubei, China). Peanut protein powder was purchased from Qingdao Changshou Co., Ltd. (Shandong, China). Soy protein isolate was purchased from Solae Co., Ltd. (Zhengzhou, China). Pork loin was purchased from a local market. Unless otherwise specified, reagents including odium dodecane sulfonate solution (SDS), NaOH and sodium phosphate were analytical grade.

### 3.2. Extraction of PPI from Peanut Protein Powder

Extraction of PPI was based on a previous study [74]. Briefly, peanut protein powder (50 g) was added to deionized water (100 mL). The mixture was adjusted to pH 9.0 and stirred at 150 rpm for 2 h. The supernatant was collected after centrifugation and kept for 1 h at pH 4.5 to precipitate the protein. The precipitation was collected after centrifugation and adjusted to pH 7.0. The suspension of PPI was freeze-dried. The content of PPI was 82.05 ± 0.36% (*w*/*w*) and measured by the Kjeldahl method.

### 3.3. Extraction of Arachin and Conarachin from PPI

Arachin and conarachin from the PPI was prepared according to Feng et al. [75]. The PPI was mixed with phosphate buffer (0.3 mol/L, pH 7.5) at a ratio of 2:5 (*w*/*v*). The mixture was stirred for 1 h and centrifuged at 8000 rpm for 30 min. The supernatant was cooled to 4 °C for 4 h and centrifuged again. Arachin was the precipitation gathered from the second centrifugation. The supernatant was adjusted to pH 4.5, and centrifuged at 4500 rpm for 20 min. The precipitate obtained was conarachin. Then, the samples of arachin and conarachin were freeze-dried.

### 3.4. Extraction of 7S Globulin from Soy Protein Isolate (SPI)

The 7S globulin was collected according to Nagano et al. [76] with light modification. SPI was added to deionized water at an ambient temperature and adjusted to pH 8.0. The mixture was stirred slowly and centrifuged to collect the supernatant. Sodium bisulfite was added to the supernatant and kept for 30 min. Then, the mixture was adjusted to pH 6.4 and kept at 4 °C overnight. The supernatant was collected after centrifugation and added NaCl with final concentration of 0.2 mol/L. The mixture was kept for 1 h at pH 5.0 and centrifuged to collect the supernatant. The pH of mixture was adjusted to 5.0 and centrifuged to collect the precipitation. The precipitation was mixed with little deionized water, then adjusted to pH 7.5 and freeze-dried.

### 3.5. Preparation of WSG from S. cerecisiae β-Glucan

*S. cerecisiae* β-glucan (1.5 g) was mixed with deionic water (100 mL). The snail enzyme (0.06 g) was added to the mixture and kept at 45 °C for 80 min. Then, the mixture was put in boiling water to deactivate the enzyme. The supernatant was collected after centrifugation at 5000 rpm and freeze-dried.

### 3.6. Preparation of WSG-PPI/SPI/Arachin/Conarachin/7S Globulin Conjugates

Conjugates were prepared based on a previous study [77] with light modification. WSG and PPI/arachin/conarachin with a proportion of 1:3 (*w*/*w*) were mixed with 0.02 M phosphate buffer and stirred for 2 h. The dispersion was adjusted to pH 9.0 and hydrated overnight. The dispersion was heated at 90 °C for 80 min and cooled down in an ice-water bath.

The WSG and SPI/7S globulin were mixed with a 0.02 M phosphate buffer at a proportion of 1:2 (*w*/*w*) and stirred for 2 h. The mixture was adjusted to pH 9.0 and hydrated overnight. Then, the dispersion was heated at 90 °C for 3 h and cooled down. The samples were dialyzed at 4 °C for 48 h and freeze-dried.

### 3.7. Functional Properties

#### 3.7.1. Emulsifying Properties

EAI and ESI were determined by Chen et al. [78]. Briefly, the conjugates’ solution obtained a WSG concentration of 9 mg/mL and protein concentration of 3 mg/mL. The solution (7 mL) and oil (3 mL) were homogenized at 10,000 rpm for 2 min. A total of 50 μL of emulsions at 0 and 10 min were added to 5 mL 0.1% sodium dodecanesulfonate solution (SDS), respectively. The absorbance of diluted emulsion at 500 nm was recorded using a UVi-2050 spectrophotometer. The EAI and ESI were calculated by the following Equations:$${\mathrm{EAI}\;(m}^{2}/g)=\frac{2\times 2.303\times A_{0}\times D}{C\times \Phi \times L\times 10^{4}}$$
$$\mathrm{ESI}\;(\min)=\frac{A_{0}\times 10}{A_{0}-A_{10}}$$
where A_0_ and A_10_ were the absorbance of the diluted emulsion at 0 and 10 min, respectively. D, C, L, and Φ were the dilution factor (100), the concentration of protein, the width of light length, and proportion of oil phase, respectively.

#### 3.7.2. Wettability Measurement

The three-phase contact angle of WSG and its conjugates was measured according to Yang et al. [79]. The solid powder of WSG and its conjugates was compressed into pellets by a tablet press, then a droplet of water was dropped on the surface of the pellets through a high-precision injector, and the picture of contact interface was taken when the contact occurred for 10 s. The contact angle of the samples was analyzed by software.

#### 3.7.3. Thermogravimetric Assay

The thermal stability of samples was measured by a thermogravimetric analyzer (Q50, Tokyo, Japan). The weight of the samples was about 6 mg. The determination conditions were as follows: the heating program was 30–500 °C, the heating rate was 15 °C/min and the nitrogen flow rate was 50 mL/min.

#### 3.7.4. Rheological Properties

Rheological properties of WSG, WSG-PPI and WSG-SPI conjugates were studied using a DHR-2 rheometer (TA Instruments, New Castle, DE, USA). The concentration of all samples was 40 mg/mL. The flow and dynamic rheological tests were determined using a 40 mm parallel plate with a 1 mm gap. The temperature was 25 °C.

#### 3.7.5. Particle Size

The particle size distribution of samples was determined by a Matersize-3000 (Malvern Instruments Ltd., Worcestershire, UK). The refractive index of the continuous phase and dispersed phased was 1.330 and 1.375, respectively.

### 3.8. Structural Characterization

#### 3.8.1. FT-IR

WSG, WSG-PPI and WSG-SPI conjugates were mixed with KBr and pressed into a transparent sheet. The parameter was set to a resolution of 4 cm^−1^ and 64 scans in the range of 400–4000 cm^−1^.

#### 3.8.2. CD

The CD spectrum was obtained by a JASCO spectropolarimeter model J-1500 (JASCO, Tokyo, Japan). Each CD spectrum was performed at a scanning speed of 100 nm/min. The wavelength range was 190–250 nm with the bandwidth set at 0.1 nm. The deionized water was used as a solvent for all the samples.

#### 3.8.3. SEM

Samples were stuck to a carbon conductive tab and sprayed with gold. The morphological properties of the samples were observed by a scanning electron microscope at 12.5 kV voltages.

#### 3.8.4. Cryo-Scanning Electron Microscope (Cryo-SEM)

The samples were formulated as a 3 mg/mL solution. Then, a droplet of solution was loaded on the cryo-specimen holder and cryo-fixed in slush nitrogen (−210 °C). After freezing, the samples were fractured in the fracture chamber. Then, the cross-section was sublimed at −90 °C and deposited, and the samples were viewed in the cryo-SEM (FEI Helios NanoLab G3 UC, State of Oregon, USA).

#### 3.8.5. AFM

The method of AFM was referred to in a previous study [80]. The samples were prepared in 50 μg/mL solution and stirred at 80 °C for 4 h. A total of 10 μL of each solution was dropped on the fresh mica, and naturally aired.

### 3.9. Application of Soluble β-Glucan-PPI Conjugation

#### 3.9.1. Preparation of Low-Fat Sausage

The formulation of sausage was showed at Table 1. The lean pork was pickled with 2% salt for 24–48 h. Then, the lean pork was mixed with ice, WSG or WSG-PPI/SPI conjugates and chopped for 2 min at high speed. The fat was added to the mixture and chopped for 5 min (total time 7 min and final temperature <12 °C). Then, the batter was stuffed into collagen casings and vacuum-packed in polyethylene bags; the sausage was cooked at 80 °C for 45 min.

#### 3.9.2. Cooking Yield of Sausage

The surface of sample was dry and the exudate was removed after the sample was cooked. The quality of sausage was measured before and after cooking.
$$\mathrm{Cooking}\;\mathrm{yield}\;(\%)=\frac{\mathrm{final}\;\mathrm{weight}\;\mathrm{of}\;\mathrm{sausage-weight}\;\mathrm{of}\;\mathrm{casing}}{\mathrm{initial}\;\mathrm{weight}\;\mathrm{of}\;\mathrm{sausage-weight}\;\mathrm{of}\;\mathrm{casing}}\times 100$$

#### 3.9.3. Color Measurement

The sausage was cut into 3 mm slide. The model chromameter was used to determine the color of sausage among different treatments for each of the 6 replications. The L*, a* and b* values were determined.

#### 3.9.4. Texture

The texture profile analysis (TPA) of the samples was measured by a TA-xT2i Texture Analyzer (Stable Micro Systems, UK), according to a method by Pires et al. [81]. The samples were divided into cylinders with a height of 2 cm after stripping the casing. The cores were compressed into two consecutive cycles of 50% using a P-36R probe. The instrument settings were as follows: pre-test speed of 2 mm/s, test speed of 1 mm/s and the post-test speed of 2 mm/s. The parameters of hardness (g), springiness, cohesiveness and chewiness were studied.

### 3.10. Statistical Analysis

All experiments were replicated 3 times, and the data were analyzed by a variance analysis (ANOVA) to determine the differences (*p* < 0.05) using SPSS 26. The graphs were made by Origin 8.5.

## 4. Conclusions

In this study, WSG-PPI and WSG-SPI conjugates were formed in the specified conditions. A Maillard reaction occurred between amino and carbonyl groups in WSG and PPI or SPI, resulting in the consumption of some functional groups and the appearance of new groups in the conjugates. In the process of the graft, the formation of WSG-PPI and WSG-SPI conjugates was observed by SEM, AFM and Cryo-SEM. The functional properties and structure of WSG-PPI and WSG-SPI conjugates changed compared to WSG. WSG-PPI and WSG-SPI demonstrated a better emulsifying property and a higher thermal stability than WSG, which was because the polysaccharide was presumably the enhanced steric stabilization provided by the bulky hydrophilic moiety. In addition, the WSG-PPI and WSG-SPI conjugates replaced part of the fat in the low-fat sausages. The cooking yield and water-holding capability improved compared to the normal fat sausage, and the hardness, springiness, chewiness and cohesiveness of sausages replaced WSG-PPI instead of 20% fat, and were more in line with the requirements of commercially available products. The range of application for the conjugates was expended.

## Figures and Tables

**Figure 1 molecules-27-03037-f001:**
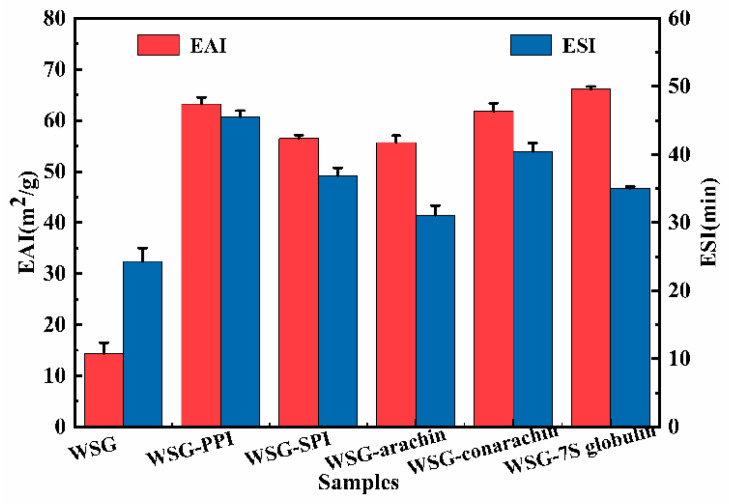
Emulsifying properties of WSG and its conjugates.

**Figure 2 molecules-27-03037-f002:**
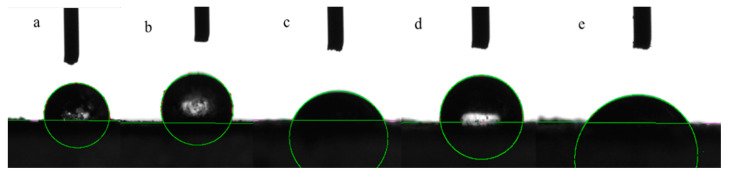
Contact angle of WSG and its conjugates: (**a**) WSG; (**b**) SPI; (**c**) PPI; (**d**) WSG-SPI conjugates; (**e**) WSG-PPI conjugates.

**Figure 3 molecules-27-03037-f003:**
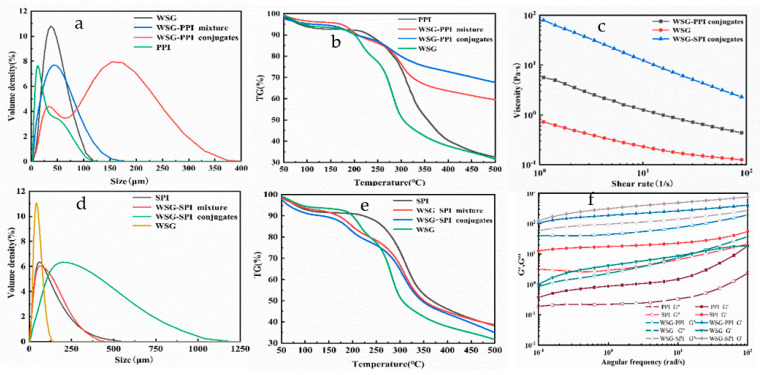
Functional properties of WSG and its conjugates: (**a**) particle size of WSG-PPI conjugates; (**b**) TGA of WSG-PPI conjugates; (**c**) rheological properties of WSG-PPI conjugates; (**d**) particle size of WSG-SPI conjugates; (**e**) TGA of WSG-SPI conjugates; (**f**) rheological of WSG-SPI conjugates.

**Figure 4 molecules-27-03037-f004:**
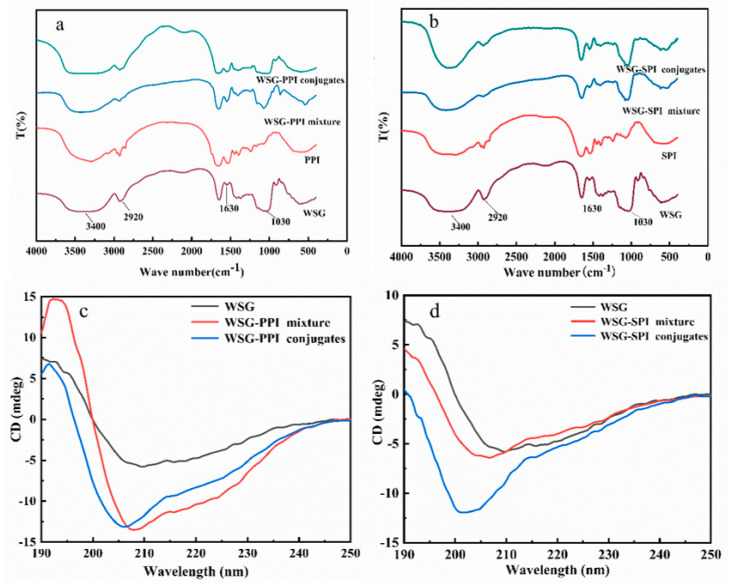
FT-IR spectra and CD spectra of WSG and its conjugates. (**a**) FT-IR spectra of WSG-PPI conjugates; (**b**) FT-IR spectra of WSG-SPI conjugates; (**c**) CD spectra of WSG-PPI conjugate; (**d**) CD spectra of WSG-SPI conjugates.

**Figure 5 molecules-27-03037-f005:**
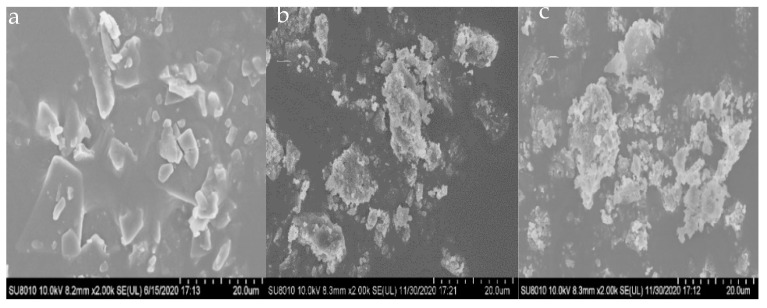
The SEM graphs of WSG and its conjugates: (**a**) WSG; (**b**) WSG-PPI conjugates; (**c**) WSG-SPI conjugates.

**Figure 6 molecules-27-03037-f006:**
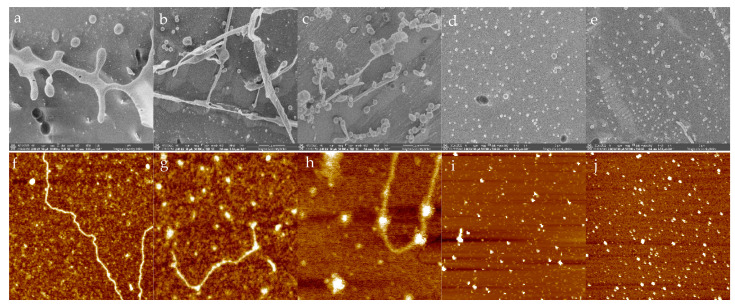
Cryo-SEM and AFM graphs of WSG and its conjugates: (**a**) Cryo-SEM graph of WSG; (**b**) Cryo-SEM graph of WSG-PPI conjugates; (**c**) Cryo-SEM graph of WSG-SPI conjugates; (**d**) Cryo-SEM graph of PPI; (**e**) Cryo-SEM graph of SPI; (**f**) AFM graph of WSG; (**g**) AFM graph of WSG-PPI conjugates; (**h**) AFM graph of WSG-SPI conjugates; (**i**) AFM graph of PPI; (**j**) AFM graph of SPI.

**Figure 7 molecules-27-03037-f007:**
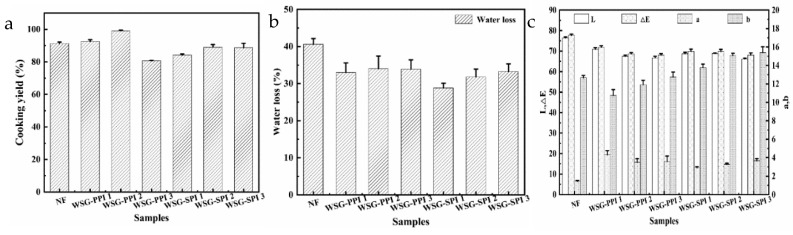
Cooking yield (**a**), water loss (**b**) and color (**c**) of low-fat sausages.

**Figure 8 molecules-27-03037-f008:**
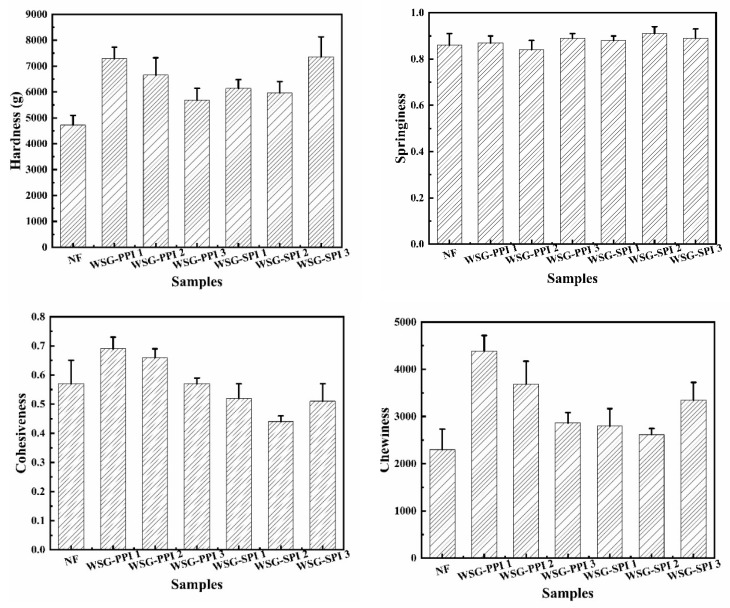
The texture of low-fat sausages.

**Table 1 molecules-27-03037-t001:** Formulations of sausage.

Treatment	Lean Pork (%)	Fat (%)	Water/Ice (%)	Salt (%)	WSG-PPI/WSG-SPI (%)
Normal	66.4	16.6	15	2	0
WSG-PPI 1	66.4	13.28	15	2	3.32
WSG-PPI 2	66.4	9.96	15	2	6.64
WSG-PPI 3	66.4	6.64	15	2	9.96
WSG-SPI 1	66.4	13.28	15	2	3.32
WSG-SPI 2	66.4	9.96	15	2	6.64
WSG-SPI 3	66.4	6.64	15	2	9.96

## Data Availability

Data are available from the corresponding author if requested.

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
