# Peer review of "Characterization of β-Glucan-Peanut Protein Isolate/Soy Protein Isolate Conjugates and Their Application on Low-Fat Sausage"

_molecules, 2022, doi:10.3390/molecules27093037_

Round 1
Reviewer 1 Report
The work is written very carefully. Every detail has been taken care of, starting with a detailed description of the methodology and material, ending with the description of the results. However, a few remarks come to my mind:
1. Please explain the WSG abbreviation. The authors do not need to read your previous works. It is not known what this abbreviation means.
2. Please more discussion! Comparisons with other works.
3. Have the authors made a sensory analysis of such products? Was it only a study an analysis? In my opinion, a few words should be added on this subject. If the authors have not done so, but intend to do so, please add the appropriate sentences.
4. Please correct the notation in Figure 3. The letters "a" "b" "c" are hard to see. Please put them at the top of the chart or in such a place that they are visible.
5. Please calculate the C* parameters; YI* in the color section.
Author Response
- Explain the WSG abbreviation
WSG was the product obtained by enzymatic hydrolysis of Saccharomyces Cerecisiae β-glucan. WSG had a reduced molecular weight and increased solubility compared to Saccharomyces Cerecisiae β-glucan. The molecular weight range of WSG was 1.40´104~2.99´106 g/mol. The solubility in water was 89.74±0.62%.
- Please more discussion! Comparisons with other works.
Comparisons with other work had been added to the article.
- Have the authors made a sensory analysis of such products? Was it only a study an analysis?
The sensory analysis of product was studied. The sausages with WSG-PPI replaced 20% fat was evaluated well. The surface of sausage was firm and smooth, and the elasticity was better. However, the color was not bright enough.
- Please correct the notation in Figure 3. The letters "a" "b" "c" are hard to see. Please put them at the top of the chart or in such a place that they are visible.
The letters “a” “b” “c” were placed in the middle of the picture for easy visibility.

Reviewer 2 Report
The manuscript titled “Characterization of β-glucan-peanut protein isolate/soy protein 2
isolate conjugates and their application on low-fat sausage” is written well and scientifically sound. It can be accepted for publication in Molecules after minor revision
Page 1 Line 44, In Introduction section, authors should discuss and include some literature about sausages
Page 1 Line 45, In Introduction section, authors should discuss and include some literature about other fat replacers in sausages
Page 8-9, mention the particle size of the glucan, PPI, SPI
Page 9, Line 301Provide temperature for rheological measurements
In Reference list, formatting is required
Author Response
- Page 1 Line 44, In Introduction section, authors should discuss and include some literature about sausages Page 1 Line 45, In Introduction section, authors should discuss and include some literature about other fat replacers in sausages
Sausage was one of popular meat products in the word, which had high content of fat and unhealthy fatty acid profile. However, with the improvement of quality of life, people pay more and more attention to the relationship between diet and health. Consumers’ purchasing habits were changing, and low-fat food became more and more popular. Fat plays an important role in maintaining the sensory characteristics such as texture and juiciness of meat product. The direct reduction of fat may lead to the poor quality of sausage. Since, many ingredients had been studied as fat replacer to decline fat levels without compromising sausage quality. Pork skin-based emulsion gels elaborated with canola oil, bamboo fiber and inulin were promising alternatives to replace pork back fat in low-fat sausages. Emulsion gels prepared with carrageenan and emulsion stabilized by zein/carboxymethyl dextrin also were served as fat substitute in sausages, which could enhance hardness and viscoelasticity of pork sausage. The addition of 15% chicken skin and wheat fiber mixture or pineapple dietary fibres to sausage could significantly reduce cooking yield. The effect of pea protein isolate, pea low moisture extrudate and pea high moisture extrudate replacing 20% pork meat were compared, there are no significant difference between sausage with PPI and normal sausage, sausage with pea extrudate had softer bite and significant color changes.
This part of content had been modified in the article.
- Page 8-9, mention the particle size of the glucan, PPI, SPI
The mean particle of WSG, WSG-PPI and WSG-SPI was 38.25±0.17 μm, 50.07± 0.76 μm and 104.89±3.79 μm.
- Page 9, Line 301Provide temperature for rheological measurements
The temperature of rheological measurement was 25 ℃

Reviewer 3 Report
First of all, This manuscript discusses an interesting point of science, which connects chemistry and biochemistry.
1 - Could authors please improve the discussion section by write it more clear and manifested the result impacts.
2 - please rewrite the conclusion to be more attractive for readers.
Author Response
- Could authors please improve the discussion section by write it more clear and manifested the result impacts.
The discussion section had been revised in the article. Comparisons with other work had been added in the article.
- please rewrite the conclusion to be more attractive for readers.
In this study, WSG-PPI and WSG-SPI conjugates were formed in the specified conditions. Maillard reaction occurred between amino and carbonyl groups in WSG and PPI or SPI, resulting in consumption of some functional groups and the appearance of new groups in the conjugates. In the process of graft, the formation of WSG-PPI and WSG-SPI conjugates was observed by SEM, AFM and Cryo-SEM. The functional properties and structure of WSG-PPI and WSG-SPI conjugates changed compared to WSG. WSG-PPI and WSG-SPI showed better emulsifying property, higher thermal stability than WSG, Which was because the polysaccharide was presumably the enhanced steric stabilization provided by the bulky hydrophilic moiety. In addition, WSG-PPI and WSG-SPI conjugates replaced part of fat to prepare low-fat sausages. The cooking yield and water-holding capability improved compared to normal fat sausage, and the hardness, springiness, chewiness and cohesiveness of sausages replaced WSG-PPI instead of 20% fat were more in line with the requirements of commercially available products. The range of application for conjugates was expended.
The part of content had added to the article.

Round 2
Reviewer 1 Report
All corrections are included. I accept the Author's answers.